# Journals with open discussion forums are excellent educational resources for peer review training exercises

Nadine Borduas-Dedekind[1], Karen C. Short[2], Samuel P. Carlson[3]

[1]Department of Chemistry, University of British-Columbia, Vancouver, V6T 1Z1, Canada
[2]USDA Forest Service, Missoula Fire Sciences Laboratory, Missoula, Montana 59808, USA
[3]Department of Land Resources and Environmental Science, Montana State University, Bozeman 59717 USA

*Correspondence to*: Nadine Borduas-Dedekind (borduas@chem.ubc.ca)

**Abstract.** Journals with open discussion forums lend themselves well for peer review exercises to train early career scientists. Earth System Science Data (ESSD) is an open-access journal for the publication of interdisciplinary datasets and articles, and is thus an example of an educational resource in the peer review process. We offer our experiences in peer review training with manuscripts submitted to ESSD, and we do so from the disparate perspectives of workshop instructor, student, and author. We then provide recommendations for the structure of a peer review workshop. We seek to promote the use of open discussion forums, including ESSD, for educational purposes, as they can provide mutual benefits to trainees, authors, reviewers, and editors.

**Short summary.** This article describes the use of the open discussion manuscript review process as an educational exercise for early career scientists.

**Main text.**

## 1. Introduction

The peer review process is an integral part of the scientific endeavour, yet most reviewers have no formal training. The learning process may have occurred by observing what reviewers write from experiences as authors or by advice from an advisor. There are resources available from publishers and scientific associations, such as Nature's Focus on Peer Review Masterclass,[1] American Chemical Society's Reviewer Lab,[2] and Wiley's Peer Review Training.[3] There are also published articles by researchers describing strategies and tips, like "Learning the Rope of Peer Reviewing",[4] "How to Write a Thorough Review",[5] "Refereeing Template: A Guide to Writing an Effective Peer Review",[6] and "The Golden Rule of Reviewing".[7] Gratifyingly, there is a growing number of outlets to help recognize the reviewers' behind-the-scenes contributions to the peer review process, such as Publons (now Web of Science), and reviewer awards by journals. These resources are great, but structured implementation of these tips and templates are required to train early career scientists.

Here, we describe a framework to apply this peer reviewing advice into a workshop for trainees. For instance, instructors can run peer reviewer training workshops within their groups or classroom to provide formal schooling in this important process. Research outlets like Earth System Science Data (ESSD), which is an open access, interactive peer reviewed journal for the publication of interdisciplinary data in the advancement of earth science, lend themselves particularly well to teaching the peer review process. Manuscripts are often extensive, and therefore different sections and dataset components can be delegated to different trainees to review. This exercise can lead to a thorough review mutually beneficial to trainees, reviewers, editors, and authors.

Authors of data publications benefit from rigorous peer review, especially in an open access, interactive forum like that of ESSD. Published datasets are intended to be used widely, and interactions with potential users help ensure the products are research- and application-ready. The interactive public discussion style of peer review can provide valuable end-user documentation beyond what is included in the final data paper or product metadata. However, a large earth science dataset may be challenging to review adequately within the typical time constraints of a publication outlet.

Since ESSD datasets and articles can be extensive, the reviewer benefits from having a team of trainees evaluate the data and the manuscript. This process ensures a high quality and thorough review, beyond what one senior reviewer

could produce. Editors can help facilitate the communication between the authors and the review team of trainees. If
this peer review training practice becomes more common, editors can start relying on these group exercises as
regular reviewers.
We, the authors of this paper, have collective experience with manuscripts published as preprints in ESSD that have
been used in peer review training exercises, and we share here our experiences. In the following sections we –
Nadine Borduas-Dedekind (NBD), Samuel Carlson (SC), and Karen Short (KS) – describe these experiences from
the perspective of instructor, student trainee, and author, respectively. We then offer recommendations for others
interested in using open discussion publishing forums for peer review training exercises.

## 2.   Personal perspectives

**Perspective from a workshop instructor (NBD)**

I am an assistant professor and my research group is composed of young researchers new to the peer review process.
To help provide transparency to the process of publishing research, I ran a workshop within my research group (2
PhD, 3 MSc and 2 BSc students) using an ESSD manuscript with a large suite of instrument data[8]. The students
were enthusiastic about participating, and I was particularly impressed with the quality of the review we wrote as a
group. During the review, the students took leadership in evaluating the data, checking databases and code, and
asking questions about the operation of different instruments. At the end of the process, our group review was
substantially more thorough than a review I could have written on my own. In addition, we included all our names
on the review to ensure the students also received credit. The authors' and the editor's feedback was excellent.
Following our posted reviewer comment, we communicated with the authors and shared the presentation of the
overview of the sections of the manuscript. As importantly, the students appreciated the behind-the-scenes look at
how a peer review was conducted. I plan to run this exercise again, either using manuscripts I receive for review, or
articles posted in open discussion forums. I recommend that authors, reviewers, editors, and readers consider this
peer reviewing practice to help train the next generation of reviewers.

**Perspective from a student (SC)**

I gained my first review experience as a participant in a collaborative student review of an ESSD manuscript. There
were approximately 10 students who participated in a one-credit special-topics class convened for this purpose. It
was instructive to learn to develop constructive criticism of a dataset and of the methods under review. For me, this
review process was the first time I had formed my own perspective on the quality and validity of data, methods, or
findings, rather than treating all scientific products as beyond reproach. This experience was a key learning
milestone in growing into an independent scientist. Contributing to the review thus pushed me to consider
assumptions incorporated in the dataset and methods. At the end of the course, the students selected a leader who
posted the reviewer comment on the open discussion forum in their name. Overall, I benefited from the opportunity
to participate in the process of science, to test my knowledge of earth science and statistical tools, and to practice
creative thinking and technical problem solving.

**Perspective from an author (KS)**

As an author of several large geospatial data publications, I have found the group-review assignment capable of
providing considerably more discussion than a single-party review within the allotted time. My initial ESSD
submission[9] was reviewed by a class of graduate students over a six week period. As a class assignment, the time
was clearly spent putting the dataset of over 1.6 million records through its paces. Feedback included thoughtful
comments on topics like data format, accessibility, quality control, and utility that I was able to respond to at length
in the interactive comment process. In contrast to typical peer reviewers, who tend to be selected because they are
inordinately familiar with the subject matter and data under consideration and therefore tend to keep their reviews
relatively "high level," the early career scientist training exercise prompted me to respond in detail to specific
questions concerning data quality and to provide usage notes that would benefit the broader user community. From
an author's perspective, I recommend having a look at published discussions[8,9] from these peer-review trainee
exercises and how they led to a high quality review of a data paper.

## 3.   Peer-reviewing training workshop

**Recommendations for training in peer reviewing**

We reflect on our respective experience as an instructor, trainee, and author to offer recommendations for a
workshop using open discussion forums to provide peer-review guidance for early career scientists. The workshop

could be embedded into a senior undergraduate or graduate course and count towards credit, or conducted within a research group. The workshop would be suited for a group of 20 participants or less to ensure adequate time for discussion and feedback. The instructor chooses a recently posted discussion paper and plans 3-4 group interactions around the manuscript. The goal of the primary exercise is to submit an open review comment reflecting the concerted efforts of the students and compiled by the instructor (who has an account with the open discussion journal). Throughout the workshop, the students read the manuscript and come together to brainstorm on the merits – or lack thereof – of the science (and data products) presented. We recommend that the instructor provides different tasks for which the trainees can volunteer. Examples of tasks related to peer-reviewing for ESSD include considerations of data accessibility, data organization, uncertainties, instruments, clarity of the writing, and recency/relevance of references. Students are then responsible individually or in smaller groups to explore sections of the manuscript and generate questions about the data, the data visualization, the data interpretation, etc. One session is then dedicated to presenting these questions to the group and attempting to answer collectively. When answers cannot be generated within the group, then these questions can be included in the reviewer document with actionable recommendation to the authors. The instructor is then responsible for the final submission of the open discussion review.

We can also recommend an additional session within the workshop where students are asked to develop potential applications of the data relevant to their interests. This element goes beyond the fundamental components of dataset review and focuses on developing students' creativity, as well as their technical abilities and understanding of statistical methods and other analytics. Consideration of potential applications, even as a proof-of-concept, can also encourage closer examination of the precision, accuracy, or quality control of the dataset and manuscript under review.

The outcomes of the workshop are for early-career scientists to learn how to ask critical questions, how to formulate suggestions for improvement using a teaching tone, and how to summarize a research article. In sum, the goals are to take part in the peer-review process, to learn about the iterative process of the scientific method and to appreciate the value of constructive criticism.

### 4. Concluding remarks
**Call to use open-discussion forums for peer-review training**
There is an intrinsic benefit when experienced scientists are investing in the future of the peer review process. If all reviewers go through a training program first, then we collectively raise the bar of the quality of the peer review process. Overall, the exposure to both the review process and the concept of openly shared, quality-assured data is important in training the next generation of scientists as well as promoting critical thinking among our trainees. We see a win-win situation for the trainee and the author involved. The concept of open data is necessary to advance knowledge more effectively and participating in all aspects of the open data review process – as a reviewer, student trainee, and author – ensures continued high-quality datasets available in ESSD and other science products.

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
