# Peer review of "Journals with open discussion forums are excellent educational resources for peer review training exercises"

_Earth System Science Data, 2022_

## Author Comment (AC1)

ESSD reviewer responses:

Reviewer 1:

This editorial provides a short collection of anecdotes that speak to the benefits of open discussion forums in providing training peer-review for early career scientists (ECS). These benefits are experienced by students providing peer review and also the authors who are receiving the peer-review.

I was particularly excited to read this manuscript. I am an editor at another Copernicus journal and have published on peer-review and held short courses similar to the one described here. I am over-averagely interested in this topic and very much welcome this contribution.

Very glad to hear about the reviewer's interest in this important topic!

**Here are some ideas for improvements and edits:**

(Abstract and paper organization) I would suggest renaming the 3 main parts of the story. Maybe split the descriptions between the author receiving and the student providing the peer review. Start with a description of the framework for the course itself.

1. A framework for a peer-review course
2. Experiences from an author
3. Experiences from a student

The reviewer has excellent ideas for restructuring the paper. We agreed that the flow could be improved. We've gone ahead and restructured our manuscript into 4 sections: Introduction, personal perspectives (which is now subdivided into first-person experiences from an instructor, a student and an author), recommendations for workshop and concluding remarks.

The edited structure flows much better and we now present recommendations of how to conduct a training exercise, instead of relating past experiences. We thank the reviewer for stimulating this restructuring.

(Line 8 and in general) I think the scope of this paper could be widened to have a (potentially) much bigger impact. Indeed ESSD is the journal in question, but actually the same ideas can be applied to any journal with an open discussion forum. I would recommend that the authors change the framing so that it includes this, but also clearly state that they use ESSD as an example. At the risk of putting words in the authors mouths, the Abstract could possibly start with something like "Journals with open discussion fora lend themselves for student peer-review exercises and training. ESSD is a good example of this, which is an open access journal for the publication of interdisciplinary datasets and articles".

Agreed! We've edited the introduction to broaden the scope beyond ESSD only, and we now relate more generally to open discussion forum style publications.

We've also edited the title to reflect this broader scope.

It now reads, "*Journals with open discussion forums are excellent educational resources for peer-review training exercises*"

(Line 20-29) I think it would be nice to tip one's hat to the many articles published on the need for training in peer-review and others who have published on how to peer review. There are many such articles out there, which would help to provide a stronger foundation for the authors claims in the first paragraph.

Agreed. We've added additional references:

(5)      Stiller-Reeve, M. How to Write a Thorough Peer Review. *Nature* **2018**. https://doi.org/10.1038/d41586-018-06991-0.
(6)      Berlinguette, C.; Gabor, N.; Surendranath, Y. "Refereeing Template": A Guide to Writing an Effective Peer Review. ChemRxiv June 8, 2021. https://doi.org/10.26434/chemrxiv.14723481.v1.
(7)      McPeek, M. A.; DeAngelis, D. L.; Shaw, R. G.; Moore, A. J.; Rausher, M. D.; Strong, D. R.; Ellison, A. M.; Barrett, L.; Rieseberg, L.; Breed, M. D.; Sullivan, J.; Osenberg, C. W.; Holyoak, M.; Elgar, M. A. The Golden Rule of Reviewing. *Am. Nat.* **2009**, *173* (5), E155–E158. https://doi.org/10.1086/598847.

We'd be happy to include other references that the reviewer might have also had in mind.

(Line 33) Also, the fact that each Copernicus journal has an open discussion forum, means that anyone with a registered account can provide a comment. Everyone's feedback is valid in an open peer-review process. In this sense, it is technically not necessary to contact the editor and author first. I would suggest that the authors change this in the text to say that anyone can actually provide a review, but we contacted the editor and author as a common courtesy. When I have held such courses before then we start the comments with a short description of how the review came about and the backgrounds and experience of the folks involved.

Agreed. The reviewer's point is exactly right. In principle, one does not have to wait until a review comes to them, but instead can use a discussion paper as the training material. This idea also helps address any timing issues (within a recurring course for example), since there are always papers in the discussion forum ready to be discussed/reviewed this this kind of peer-review exercise.

Now that we've restructured our discussion to present recommendations for a peer reviewing workshop, instead of relating our specific experiences, we've followed the reviewer's comment and stated explicitly that a instructor can submit a review at any time.

The text now reads, *"The goal of the primary exercise is to submit an open-review comment reflecting the concerted efforts of the students and compiled by the instructor (who has an account with the open discussion journal).*

The first few lines (63-69) of the section "Author receiving a peer-review report from a team of students" would be more appropriately positioned in the Introduction. I would like to hear a few more details in this section about what kind of feedback the author received and potentially how the feedback varied (or not) from a standard peer-reviewer.

We've now restructured our manuscript and expanded our introduction. We provide the feedback for the instructor, student and author in more details and follow these perspectives

with recommendations on a workshop. We think this restructuring also helps address this comment.

I have an issue with the use of "I" and "us" and "we" in the text, which I am sure you can find a solution for. It gets a little confusing when there are 3 authors and the first-person pronouns relate to different authors throughout the text. One suggestion could be to provide the authors name in brackets in each of the subheadings. For example, "Student providing a peer-review report (by Samuel Carlsen)". Or find another rhetorical move that tackles this issue. Either way, I feel this needs to be resolved.

We agree with the reviewer's criticism and have struggled with addressing the flow of the text. We think that now with the restructuring of the sections, we can provide a clearer flow and tone. We therefore use "we" throughout the text, and as the reviewer suggested, use "I" within specific sections with our names associated.

Finally, the authors do a good job at presenting the positive aspect of such training exercises. However, I think it could be a healthy to ponder potential pit-falls in training processes such as this. Are there any?

The reviewer makes a fair comment. We're not suggesting that all reviews be by students. A combination of expert review and student review could be ideal. There might also be a longer timeline to a student peer review. On a related note, we consider that the students are likely going to be the future users of ESSD datasets and now specifically emphasize this point.

The text now reads, "*We can also recommend an additional session within the workshop where students are asked to develop potential applications of the data relevant to their interests. This element goes beyond the fundamental components of dataset review and focuses on their students' creativity, as well as their technical abilities and understanding of statistical methods and other analytics. Consideration of potential applications, even as a proof-of-concept, can also encourage closer examination of the precision, accuracy, or quality control of the review dataset.*"

This exercise is overall a win-win situation for everyone involved. The text now reads, "*We see a win-win situation for the trainee and the author involved.*"

I very much welcome this contribution to the literature on peer-review training. It provides a citable resource for me to justify many of my own practices, which I appreciate. With some easy changes, I believe this editorial could have a wider impact than just to the readership of ESSD (which it seemingly targets at present).

This reviewer's enthusiasm and feedback is very much appreciated. With the critical feedback from this reviewer, our manuscript has significantly improved. Thank you!

---

## Author Comment (AC2)

Reviewer 2:

This editorial discusses a peer-reviewing training exercise for students and young scientists using ESSD platform. As many young scientists didn't get any peer-review training before they start reviewing scientific manuscripts, it is a great idea to take advantage of open access and interactive platforms such as ESSD to train them under the mentorship from experienced scientists. Such practice can be very valuable not only to young scientists but also the science community. I only have minor comments and suggestions for the authors to consider.

As pointed out by the other reviewer, the authors should widen the scope and potential impact of this practice. Particularly consider rephrasing 'student training exercise' as 'early-career scientist training exercise'. Also, although this exercise used an ESSD manuscript, it can be applied to other interactive journals for future exercise, the authors should emphasize this point.

Agreed. We changed the term student to early career scientist when speaking of the broader context and of the benefits of the exercise. We did keep the word student when referring to our workshop description and to the student perspective, as the term more precisely describes the targeted audience during the exercises we are describing.

This practice requires mentorship from experienced scientists who are normally very busy. Any suggestions on how to encourage senior scientists to do so? How should ESSD (or other similar journals) and institutions support such training exercise? Any improvements that the authors will do for future training exercise?

This comment is a fair point. We don't have obvious solutions but brainstorm below:

1. In the context of a course, the instructor and the student obtain credit for teaching and taking the course, respectively.
2. There is an intrinsic benefit when experienced scientists are investing in the future of the peer-review process. If all reviewers must go through a training program first, then we raise the bar of the quality of reviewing.
3. Journals could incentivise these types of training by requiring that new reviewers participate in a short online course before accepting to review for the first time. This training can be a short video followed by a quiz. However, reviewers are already hard to come by, and we can imagine that editors might oppose this process as it might slow the onboarding of new reviewers. Either way, there could be future discussions on this point in order to raise the effectiveness of the peer-review process.

Line 43: 'one of us' to 'one of the authors'

We've tried to rehaul the use of "I" and "we" throughout the article to avoid authors' voice confusion. We think that now with the restructuring of the sections, we can provide a clearer flow and tone. We therefore use "we" throughout the text, and as the reviewer suggested, use "I" within specific sections with our names associated.

Line 44: I was confused when I read ' the goal of this workshop' when no other description of the workshop is given. Perhaps rewrite the sentence into something like 'The authors organized a peer-review workshop for early-career scientists, the goal was xxx '

Fair point. In our restructuring of the article, we now describe the workshop as well as the course in more detail in a separate section. We've also revised the manuscript to focus on recommendations of a workshop instead of what had been previously done.

Lines 47-49: how many undergraduate and graduate (Master and PhD) students? How were they assigned to work on different sections of the manuscript (random, or students' interest)?

In the instructor experience described in the manuscript, there were 2 PhD, 3 MSc and 2 BSc students in the (now explicitly stated in the manuscript). The assignment in this case was on a volunteer basis.

In the student experience described in the manuscript, the instructor identified specific aspects or sections of the manuscript for all of the students to review each week. Additionally, students considered applications for the data based on their individual interests. This group consisted of about 10 students, a majority of which were graduate students. This information has been added to the manuscript.

Lines 60-61: are there any recognitions for the students, e.g. their names mentioned in the review reports?

In the case of NBD's experience, yes, the names of all student participants were mentioned in the reviewer report which is publicly posted on the discussion tab of the article (now mentioned explicitly). In the case of the course, only one student name was posted in the comment.